# MURKA: Multi-Reward Reinforcement Learning with Knowledge Alignment for Optimization Tasks

**Wantong Xie[1], Yi-Xiang Hu[2], Jieyang Xu[2], Feng Wu[2]\*, Xiang-Yang Li[2]\***

[1]Institute of Advanced Technology, University of Science and Technology of China
[2]School of Computer Science and Technology, University of Science and Technology of China
{wantongxie,yixianghu,jieyangxu}@mail.ustc.edu.cn
{wufeng02,xiangyangli}@ustc.edu.cn

## Abstract

Optimization plays a central role in Operations Research (OR) and numerous industrial applications, yet automating the end-to-end process of translating natural language descriptions into executable optimization programs remains a formidable challenge. While recent efforts have applied Large Language Models (LLMs) to this task, existing approaches are hindered by high inference costs, limited robustness across domains, and weak verification mechanisms. In this work, we propose MURKA, a reinforcement learning and knowledge distillation-based framework that enhances LLM-driven optimization modeling via collaborative agent alignment. MURKA orchestrates three specialized agents—Extractor, Solver, and Checker—to achieve accurate problem understanding, robust formulation, and verifiable execution. The Extractor is trained using group relative policy optimization with a composite reward function that incorporates semantic correctness and execution fidelity. The Solver benefits from knowledge distillation from a powerful teacher model, yielding structurally valid and executable formulations in AMPL. The Checker iteratively verifies solution correctness via solver feedback. We validate MURKA's generalizability through extensive experiments across diverse OR benchmarks, demonstrating its robustness and scalability. Experimental results on eight diverse OR benchmarks, including NLP4LP, ComplexOR, and NL4Opt, demonstrate that MURKA, built on the LLaMa3-8B backbone, achieves a 5.9% absolute improvement in solution accuracy and a 5.1% increase in execution success rate compared to leading baselines. These results establish MURKA as an effective and scalable paradigm for LLM-driven optimization, with strong potential for deployment in real-world OR applications.

## 1 Introduction

Operations Research (OR) has long served as a cornerstone for solving complex decision-making problems across domains such as defense, logistics, supply chain management, and business operations [Trimborn et al., 2020]. However, the traditional OR modeling pipeline is notoriously labor-intensive, requiring substantial expertise to manually translate real-world objectives into structured mathematical formulations. This expert-driven workflow presents significant scalability challenges, especially in fast-evolving industrial environments.

With the rapid advancement of Large Language Models (LLMs), exemplified by ChatGPT-4 [Achiam et al., 2023] and DeepSeek-R1 [Guo et al., 2025], a new paradigm has emerged: using LLMs to automate the translation of natural language descriptions into optimization models. Recent

---

\*Corresponding authors.

studies [Xiao et al., 2023, Li et al., 2023, AhmadiTeshnizi et al., 2024, Zhang et al., 2024, Huang et al., 2025] have demonstrated the potential of LLMs to reduce expert dependency and accelerate OR modeling. These methods can be broadly categorized into (1) prompt-based approaches, which rely on carefully engineered prompts to guide LLMs, and (2) learning-based approaches, which fine-tune models via supervised learning or Reinforcement Learning (RL).

Despite their promise, the deployment of such methods in real-world scenarios remains limited. Three critical challenges continue to impede the practical adoption of LLMs for OR modeling: (C1) *Scalability and latency* - Large models impose high computational costs, making real-time or edge deployment infeasible, particularly in industrial systems requiring low-latency decisions; (C2) *Semantic extraction and modeling accuracy* - Existing approaches struggle to balance modeling correctness and efficiency. Manual annotation is expensive and non-scalable, while automatic extraction methods often suffer from semantic drift and ambiguity; (C3) *Cross-domain generalization* - Many LLM-based methods exhibit limited adaptability across heterogeneous optimization domains, such as manufacturing, energy, and logistics.

To address these challenges, we propose **MURKA**, a collaborative multi-agent framework that aligns LLMs with optimization tasks via RL. MURKA orchestrates three specialized agents–*Extractor*, *Solver*, and *Checker*–to systematically convert natural language problem descriptions into verifiable optimization solutions through an end-to-end, multi-stage pipeline. Our contributions can be summarized as follows:

- **Modular multi-agent framework.** To address (C1), we design a collaborative multi-agent framework that decomposes the modeling process into information extraction, model generation, and verification stages, enhancing efficiency, modularity, and interpretability.

- **Combined multi-reward RL strategy.** To tackle (C2), we propose a composite reward function that integrates format validation, constraint checking, semantic analysis, and similarity checking.

- **Cross-domain alignment via knowledge distillation.** To mitigate (C3), we distill reasoning patterns from a high-capacity teacher model into a compact solver model, producing executable AMPL code with improved structural and functional accuracy.

- **Substantial improvements over existing methods.** MURKA achieves state-of-the-art performance on eight diverse OR benchmarks, including NLP4LP, ComplexOR, and NL4Opt, with a 5.9% relative improvement in solution accuracy and a 5.1% increase in execution success rate compared to previous methods.

Together, these innovations position MURKA as a robust, scalable, and generalizable solution for LLM-driven optimization modeling, with strong applicability to a broad spectrum of real-world OR tasks.

## 2 MURKA Framework

### 2.1 Overview

Given an optimization problem described in natural language $\mathcal{Q}$, our goal is to produce its optimal solution $\mathcal{A}$, as shown in Figure 1. We realize it through a multi-agent framework, MURKA, which can be formalized as an automatic four-stage description-to-solution pipeline:

$$\Phi : \mathcal{Q} \xrightarrow{\mathcal{E}} \mathcal{I} \xrightarrow{\mathcal{S}} M \xrightarrow{\text{Optimizer}} \hat{\mathcal{A}} \xrightarrow{\mathcal{C}} \mathcal{A}, \tag{1}$$

where $M$ typically takes the linear form

$$\min_{x} \quad Z = \sum_{i \in I} a_i x_i \quad \text{s.t.} \quad \sum_{i \in I} g_{ij} x_i \leq c_j, \quad \forall j \in J, \tag{2}$$

but the framework generalises to mixed-integer and non-linear cases.

1. **Information Extraction** ($\mathcal{E}$). An *Extractor* agent parses $\mathcal{Q}$ and outputs a structured tuple $\langle \mathcal{S}, \mathcal{P}, \mathcal{V}, \mathcal{C}, \mathcal{O} \rangle$ containing sets, parameters, decision variables, constraints, and the objective function.

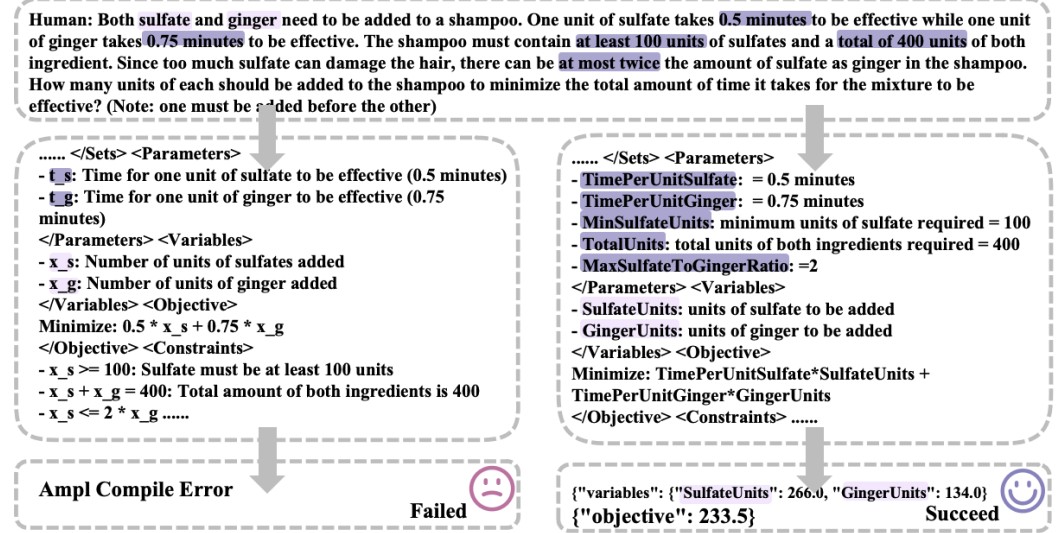

Figure 1: The impact of information quality on the solution results. The color purple represents a parameter, and violet represents a variable. The left side extracts low-quality information, leading to incorrect results, while the right side extracts high-quality information, resulting in correct outcomes.

2. **Code Generation** ($\mathcal{S}$). A Solver agent transforms $\mathcal{Q}$ and $\mathcal{I}$ into an executable AMPL model $M$ [Fourer et al., 2003], which links the problem's mathematical formulation to an external optimization engine.

3. **Numerical Optimisation**. The generated code calls the optimizer, such as Gurobi [Gurobi Optimization, LLC, 2025] solver, to obtain a candidate solution $\hat{\mathcal{A}}$ together with diagnostic information (status, duals, objective value).

4. **Iterative Verification** ($\mathcal{C}$). A *Checker* agent validates $\hat{\mathcal{A}}$ against the original specification; if infeasibilities or format errors are detected, it returns corrective feedback to the Solver, triggering another generate-solve cycle until a feasible, verified solution $\mathcal{A}$ is produced.

**Illustrative example.** For a natural-language diet problem, the Extractor retrieves the food set, nutritional constraints, and cost coefficients; the Solver converts them into the corresponding AMPL model; the numerical optimizer returns the optimal food quantities, and the Checker verifies nutritional feasibility before providing the final answer.

Therefore, as shown in Figure 2 (a), MURKA includes three important agents: *Extractor*, *Solver*, and *Checker*. They complete the entire optimization problem-solving task through collaboration. The Extractor is obtained through the training of the Group Relative Policy Optimization (GRPO) [Shao et al., 2024] RL with our specially improved and designed efficient multi-dimensional rewards (see §2.2). The Solver first synthesizes data through a deep thinking and reasoning model, and then is fine-tuned by knowledge distillation (see §2.3).

## 2.2 Extractor Alignment via Combined Reward Reinforcement Learning

The information extraction task for optimization problem formulation involves parsing complex natural language descriptions into structured components, making it well-suited for RL due to its sequential decision-making nature and the need to optimize extraction quality through iterative feedback [Pan et al., 2025]. To model this task, we adapt the GRPO algorithm. The process begins with generating multiple candidate solutions, referred to as completions, to explore diverse extractions of key problem elements. Specifically, at each training step, we sample a batch of optimization problems and generate $G$ completions (denoted $o_i$) for each problem, representing structured information extractions. Next, we compute the advantage value for each completion to guide policy updates. In the original GRPO, this relies on a reward model, which poses challenges in practical

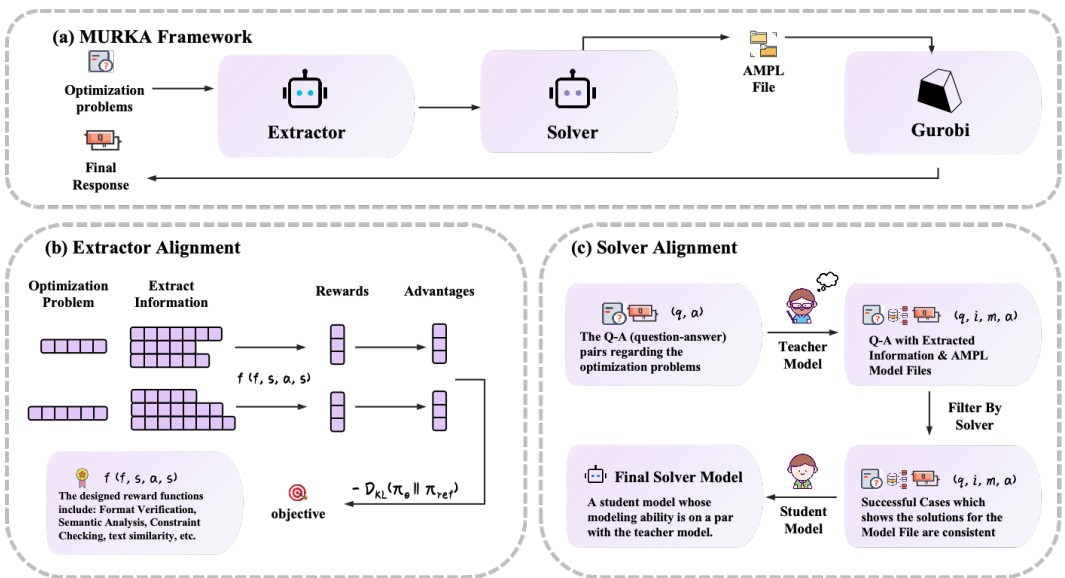

Figure 2: MURKA framework pipeline. (a) Multi-agent orchestration of Extractor-Solver-Checker agents for optimization problem solving. (b) RL training workflow for the Extractor with multi-dimensional reward signals. (c) Knowledge distillation process for Solver alignment through teacher-student model transfer.

settings due to its high data and computational demands and sensitivity to training data quality. To address these limitations, we propose an enhanced GRPO approach that replaces the reward model with a composite reward function $r(o_i)$, detailed in Section 2.4. For each of the $G$ completions, the reward is calculated using $r(o_i)$, enabling the advantage value $\hat{A}_{i,t}$ to better reflect the quality of extractions and provide more effective policy optimization. The advantage is normalized as shown in Equation 3, where $\mathrm{mean}(\mathbf{r}(\mathbf{o}))$ and $\mathrm{std}(\mathbf{r}(\mathbf{o}))$ represent the mean and standard deviation of the rewards across completions, respectively.

$$\hat{A}_{i,t} = \frac{r(o_i) - \mathrm{mean}(\mathbf{r}(\mathbf{o}))}{\mathrm{std}(\mathbf{r}(\mathbf{o}))} \tag{3}$$

In the links of estimating the KL divergence and calculating the loss, we follow the methods of the original GRPO algorithm. The KL divergence $\mathbb{D}_{KL}$ is used to measure the difference between two policies, and its definition is shown in Equation 4, where $\pi_\theta$ and $\pi_{\theta_{ref}}$ are the current policy (determined by the model parameter $\theta$) and the reference policy, respectively. $\pi_\theta\left(o_{i,t} \mid q, o_{i,<t}\right)$ represents the probability of taking the action $o_{i,t}$ at the $t$-th moment of the $i$-th sample under the policy $\pi_\theta$, given the optimization problem $q$ and the previous information extraction $o_{i,<t}$. By estimating $\mathbb{D}_{KL}$, the magnitude of the policy update is punished to ensure that the model does not deviate too much from the reference policy when updating the policy, maintaining the stability of the training process.

$$\mathbb{D}_{\mathrm{KL}}\left[\pi_\theta\|\pi_{\mathrm{ref}}\right] = \frac{\pi_{\mathrm{ref}}\left(o_{i,t} \mid q, o_{i,<t}\right)}{\pi_\theta\left(o_{i,t} \mid q, o_{i,<t}\right)} - \log \frac{\pi_{\mathrm{ref}}\left(o_{i,t} \mid q, o_{i,<t}\right)}{\pi_\theta\left(o_{i,t} \mid q, o_{i,<t}\right)} - 1 \tag{4}$$

When calculating the loss, the loss function is still determined by maximizing the advantage and considering the KL divergence penalty, and its definition is shown in Equation 5. The first term represents the scaled advantage, and the second term punishes the deviation from the reference policy through $\mathbb{D}_{KL}$. Where $\beta$ is a hyperparameter used to control the degree of KL divergence penalty, and $nograd$ means that the old policy term does not participate in the gradient calculation.

$$\mathcal{L}(\theta) = -\frac{1}{G} \sum_{i=1}^{G} \frac{1}{|o_i|} \sum_{t=1}^{|o_i|} \left[ \frac{\pi_\theta\left(o_{i,t} \mid q, o_{i,\leq t}\right)}{\left[\pi_{\theta_{old}}\left(o_{i,t} \mid q, o_{i,<t}\right)\right]_{\mathrm{no\ grad}}} \hat{A}_{i,t} - \beta \mathbb{D}_{KL}\left[\pi_\theta\|\pi_{ref}\right] \right] \tag{5}$$

---

**Algorithm 1** Comprehensive Reward Calculation

---

**Require:** Candidate response $c_i$, weights $\omega_{\text{format}}, \omega_{\text{constr}}, \omega_{\text{sem}}, \omega_{\text{sim}}$
**Ensure:** Comprehensive reward $R$
 1: Extract text content $t_i \leftarrow c_i.\text{content}$
 2: Compute format reward $R_{\text{format}}$     *(see Algorithm 2, lines 2–6)*
 3: Compute constraint reward $R_{\text{constr}}$     *(see Algorithm 2, lines 7–14)*
 4: Compute semantic reward $R_{\text{sem}}$     *(see Algorithm 3, lines 2–5)*
 5: Compute similarity reward $R_{\text{sim}}$     *(see Algorithm 3, lines 6–13)*
 6: Compute comprehensive reward:
 7:     $R \leftarrow \omega_{\text{format}} \cdot R_{\text{format}} + \omega_{\text{constr}} \cdot R_{\text{constr}} + \omega_{\text{sem}} \cdot R_{\text{sem}} + \omega_{\text{sim}} \cdot R_{\text{sim}}$
 8: **return** $R$

---

## 2.3 Solver Alignment through Knowledge Distillation

Inspired by Hinton et al. [2015], our Solver model alignment strategy employs knowledge distillation to transfer optimization modeling expertise from a high-capacity teacher model to a compact student model, as shown in Figure 2(c). We curate a diverse training dataset of 3,602 problem instances, sampled from 20% of the test set and stratified by scenarios and optimization types (Table 10, Table 11). Using teacher models like DeepSeek-R1 [Guo et al., 2025], we generate enriched outputs $(q, i, m, \alpha)$ from question-answer pairs $(q, \alpha)$, where $i$ is the extracted information and $m$ is the AMPL model. Data augmentation via prompt engineering and domain-expert input enhances diversity, while an iterative refinement process using Gurobi [Gurobi Optimization, LLC, 2025] validates and adjusts AMPL models until they match the ground-truth $\alpha$, ensuring high-quality training data.

## 2.4 Combined Reward Implementation Details

Additional symbol explanations are provided in Appendix A. Algorithm 1 integrates format validation, constraint checking, semantic analysis, and similarity checking, calculating a comprehensive reward $R$ through weighted summation to evaluate the overall quality of model-generated results. Specific implementation details for each stage are provided in Appendix B. Algorithms 2 and 3 describe our method for comprehensively evaluating model-generated results using multi-dimensional reward functions, with format validation, semantic analysis, and constraint checking as core stages to compute the final reward value.

Our methodology establishes a multi-criteria reward mechanism through formal composition of four orthogonal evaluation dimensions. Let $c_i$ denote the candidate response, the composite reward $R(c_i)$ is formulated as:

$$R(c_i) = \sum_{k \in \{\text{fmt, constr, sem, sim}\}} \omega_k R_k(c_i) \tag{6}$$

where $\omega_k$ denotes configurable weight coefficients reflecting dimensional importance. The constituent rewards are computed as follows:

**Format Verification.** This component evaluates XML structure compliance through pattern matching and positional analysis:

$$R_{\text{fmt}} = \underbrace{r \cdot \mathbb{I}(p_{\text{strict}} \sqsubseteq t_i)}_{\text{strict match}} + \underbrace{r \cdot \mathbb{I}(p_{\text{soft}} \preceq t_i)}_{\text{soft match}} \\ + \sum_{s \in S} [r \cdot \mathbb{I}(\text{pos}(s) \in \Phi_s)] - \alpha \|l_{\text{pre}} + l_{\text{post}}\|_1 \tag{7}$$

where $\Phi_s$ denotes valid positional constraints for tag $s$, and $\|\cdot\|_1$ measures redundant text length.

**Constraint Checking.** This reward evaluates mathematical constraint validity through operator-variable interaction:

$$R_{\text{constr}} = \mathbb{I}_{\text{exist}} \cdot r + \sum_{k=1}^{K} \left[ \mathbb{I}_{\text{op}_k} + \mathbb{I}_{v_k \in V \cup P} + \mathbb{I}_{\geq 0} \right] \cdot r \tag{8}$$

with $\text{clip}(R_{\text{constr}}, R_{\min}, R_{\max})$ ensuring numerical stability.

**Semantic Analysis.** Element-level analysis of objective function components:

$$R_{\text{sem}} = \text{clip}\left( \sum_{e \in E} \left[ r_1 \mathbb{I}_{\text{num}(e)} + r_2 \mathbb{I}_{e \in V \cup P} - r_3 \mathbb{I}_{\text{undef}(e)} \right], R_{\min}, R_{\max} \right) \tag{9}$$

**Similarity Checking.** This metric combines cardinality alignment and embedding-based similarity, with embeddings generated by the nomic-embed model [Nussbaum et al., 2024]:

$$\mathcal{S}(A, B) = \frac{1}{|A|} \sum_{a \in A} \max_{b \in B} \cos(\mathcal{M}_{\text{emb}}(a), \mathcal{M}_{\text{emb}}(b))$$
$$R_{\text{sim}} = \text{clip}\left( -\beta \Delta_{\text{card}} + \delta(\gamma_1 \mathcal{S}(V_{\text{aim}}, V_{\text{act}}) + \gamma_2 \mathcal{S}(P_{\text{aim}}, P_{\text{act}})), R_{\min}, R_{\max} \right) \tag{10}$$

where $\Delta_{\text{card}} = ||V_{\text{aim}}| - |V_{\text{act}}|| + ||P_{\text{aim}}| - |P_{\text{act}}||$ measures set cardinality discrepancy.

The complete reward system employs hierarchical boundary constraints and weight calibration to maintain numerical stability while preserving gradient information for policy optimization. Hyperparameters $(\alpha, \beta, \delta, \gamma_i, \omega_i)$ are grid-optimized through orthogonal experimental design.

## 3 Experiments

We conduct an extensive evaluation of MURKA to demonstrate its effectiveness in automating optimization problem modeling, comparing it against a diverse set of state-of-the-art baseline methods across multiple dimensions. Our experimental design encompasses several objectives: (1) Assessing MURKA's performance against leading LLM-assisted, multi-agent, domain-specific, and general optimization approaches; (2) Validating the contributions of its Extractor and Solver components through ablation studies; (3) Evaluating its generalization across diverse tasks beyond optimization; (4) Analyzing the impact of training hyperparameters on performance stability; (5) Exploring its scalability on smaller-scale models. Detailed implementation specifics, including experimental descriptions, prompt templates, hyperparameter configurations, and computational resources, are provided in Appendix D and Appendix E.

### 3.1 Experimental Setup

**Benchmarks.** We evaluated our method on eight diverse benchmarks—NL4Opt [Ramamonjison et al., 2023], Mamo Easy, Mamo Complex [Huang et al., 2024], NLP4LP [AhmadiTeshnizi et al., 2024], ComplexOR [Xiao et al., 2023], IndustryOR [Huang et al., 2025], OptiBench [Yang et al., 2024], and OptMATH [Lu et al., 2025]—comprising 2,224 problem instances. These benchmarks cover over 20 real-world scenarios, including Agriculture, Transportation, and Entertainment, and span seven optimization categories, such as Linear Programming, Mixed-Integer Programming, and Combinatorial Optimization. Details are provided in Appendix C.

**Baselines.** To benchmark our method, we compare against four categories of baseline approaches:

1. **LLM-Assisted Optimization Methods.** Methods integrating advanced large language models, such as GPT-4 [Achiam et al., 2023], Qwen-3 [Team, 2025] and DeepSeek-R1 [Guo et al., 2025], with the Gurobi solver [Gurobi Optimization, LLC, 2025] for optimization tasks.

2. **Multi-Agent Optimization Frameworks.** Frameworks employing collaborative multi-agent strategies, including Reflexion [Shinn et al., 2023], which enhances decision-making through verbal reinforcement learning without model fine-tuning; COE [Xiao et al., 2023], which uses a chain of 11 expert agents with forward thinking and backward reflection to solve complex operations research problems; and OptiMUS [AhmadiTeshnizi et al., 2024], which builds and solves

Table 1: The SA compared to the domain-specific optimization models. The best results among previous alignment works are marked with underline, the overall best results are marked in **bold**, and values in purple indicate that our method outperforms the previous alignment baselines.

| Method | Model | NL4Opt | Mamo Easy | Mamo Complex | Specific[†] | Micro Avg | Macro Avg |
|--------|-------|--------|-----------|--------------|-------------|-----------|-----------|
| Directly | GPT-4 | 47.3% | 66.5% | 14.6% | 22.3% | 49.19% | 37.68% |
|  | DeepSeek-R1 | 94.8% | **95.9%** | 51.2% | **45.2%** | **82.48%** | **71.78%** |
| OptMATH | Qwen2.5-7B | 94.7% | 86.5% | 51.2% | 24.4% | 75.23% | 64.20% |
|  | Qwen2.5-32B | 95.9% | 89.9% | 54.1% | 34.7% | 78.88% | 68.65% |
| ORLM | Mistral-7B | 84.4% | 81.4% | 32.0% | 27.0% | 67.56% | 56.20% |
|  | Math-7B[‡] | 86.5% | 82.2% | 37.9% | 33.0% | 70.05% | 59.90% |
|  | LLaMa3-8B | 85.7% | 82.3% | 37.4% | 38.0% | 70.42% | 60.85% |
| Ours | LLaMa3-8B | 93.5% | 95.9% | 55.6% | 37.4% | 82.18% | 70.60% |

[†] Specific represents the average of IndustryOR and OptMATH.
[‡] Math-7B represents the DeepSeek-Math-7B-Base.

linear and mixed-integer programming models from natural language using modular multi-agent collaboration.

3. **Domain-specific Customized Optimization Models.** Specialized approaches for industrial operations research and mathematical optimization, including ORLM [Huang et al., 2025], which trains open-source large language models via OR-Instruct synthetic datasets, and OptMATH [Lu et al., 2025], which generates large-scale datasets using selected seed data and back-translation to fine-tune LLMs for automated optimization modeling.

4. **General Optimization Methods.** Methods for generic optimization problems, including NL2OR [Li et al., 2024], which translates natural language into operations research models for accessible solutions, and LLMOPT [JIANG et al., 2024], a learning-based framework that defines and solves diverse optimization tasks with improved generalization.

**Metrics.** In the experiment, we use three performance metrics to comprehensively evaluate the quality of the generated code of the algorithm, namely, *Solution Accuracy (SA)*, which measures the correctness of the provided solutions, *Compilation Accuracy (CA)*, which assesses the ability of the code to compile without errors, and *Execution Rate (ER)*, which indicates the proportion of successful executions.

### 3.2 Performance Evaluation of MURKA Against Optimization Baselines

**LLM-Assisted Optimization Methods Comparison.** As shown in Figure 3a, MURKA achieves a macro-average performance of 68.61%, closely approaching DeepSeek-R1's 71.13%, despite using a significantly smaller LLaMa3-8B backbone. It is critical to note that the DeepSeek-R1-671B baseline is over 80 times larger than our model. Despite this scale disparity, MURKA demonstrates comparable and even superior performance on specific benchmarks, underscoring our framework's substantial computational efficiency and viability for practical deployment. Our selection of LLaMa3-8B was predicated on its strong, publicly verifiable performance, establishing it as a robust foundation for our experiments. However, we emphasize that MURKA's core technical contribution is its model-agnostic architecture. The framework is designed as a modular, "plug-and-play" enhancement that can be seamlessly applied to other capable base models, such as Qwen-7B or future open-source alternatives. This adaptability ensures that the demonstrated performance gains are attributable to our framework's structured approach rather than the intrinsic capabilities of a single base model.

We hypothesize that this competitive performance stems from the nature of NL-to-Optimization challenges. These tasks present two distinct hurdles: (1) *complex reasoning chains*, which require intricate, multi-step logical deductions, and (2) *robust structural extraction*, which demands precise parsing of numerical values, relationships, and constraints from dense or ambiguous text. While large models with strong general reasoning capabilities, such as DeepSeek-R1, naturally excel at the former, their generalist approach can be brittle when faced with the latter. In contrast, MURKA's Ex-

Table 2: The SA and ER compared to the general optimization methods. The best results among previous alignment works are marked with underline, and values in purple indicate that our method outperforms the previous alignment baselines.

| Metrics | ER | SA | ER | SA | ER | SA | ER | SA | ER | SA |
|---|---|---|---|---|---|---|---|---|---|---|
| Method | NL4Opt | | Mamo Easy | | Mamo Complex | | NLP4LP | | Micro Avg | |
| NL2OR† | 86.5% | 74.8% | 70.1% | 63.2% | 22.8% | 18.0% | 88.8% | 75.6% | 59.5% | 48.3% |
| LLMOPT† | 99.0% | 93.0% | 100.0% | 95.3% | 98.0% | 68.0% | 100.0% | 83.8% | 92.7% | 76.7% |
| Ours | 100.0% | 93.5% | 100.0% | 95.9% | 100.0% | 55.6% | 100.0% | 87.6% | 97.0% | 76.5% |
| Method | ComplexOR | | IndustryOR | | OptiBench | | OptMATH | | Macro Avg | |
| NL2OR† | 11.1% | 5.56% | 14.0% | 4.0% | 62.3% | 43.1% | 6.63% | 2.41% | 45.3% | 35.8% |
| LLMOPT† | 94.7% | 72.7% | 92.0% | 44.0% | 82.3% | 66.4% | 75.3% | 40.0% | 92.7% | 70.4% |
| Ours | 100.0% | 72.2% | 99.0% | 38.0% | 91.4% | 69.3% | 92.2% | 36.8% | 97.8% | 68.6% |

† NL2OR uses LLaMa3-70B as its backbone, while LLMOPT uses Qwen1.5-14B as its backbone.

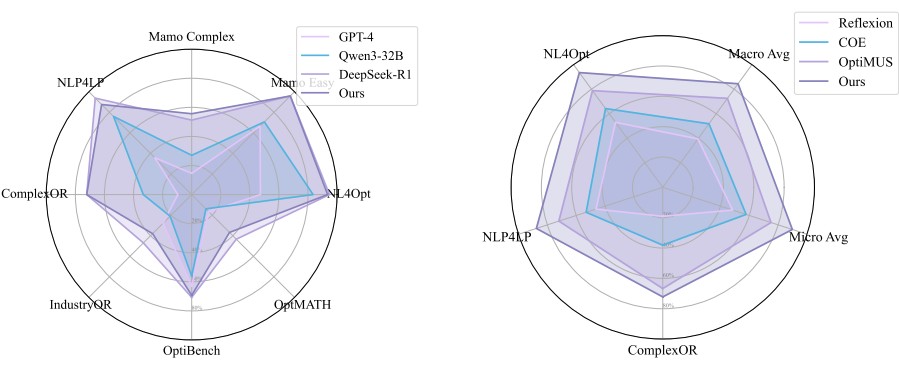

(a) LLM-Assisted Methods Comparison      (b) Multi-Agent methods Comparison

Figure 3: The SA compared to LLM-Assisted optimization methods and Multi-Agent methods.

tractor agent is specifically optimized via our multi-dimensional RL reward to achieve high-fidelity structural parsing, making it robust against the very ambiguities that can derail a general reasoning process. In essence, MURKA bridges the performance gap not by mirroring general problem-solving abilities, but by specializing in the systematic and verifiable translation of natural language into structured optimization models.

**Multi-Agent Optimization Frameworks Comparison.** Our approach delivers state-of-the-art performance across all benchmarks (Figure 3b), with absolute improvements of 14.8% and 11.93% in micro- and macro-average metrics, respectively, over prior multi-agent methods. These gains, consistent across standard and complex scenarios, highlight the strength of our structured task decomposition framework in enhancing collaborative reasoning.

**Domain-Tailored Customized Optimization Models Comparison.** Our method outperforms specialized optimization models by 3.4% in micro-average and 1.75% in macro-average metrics (Table 1). It achieves 95.9 on Mamo Easy and 55.6 on Mamo Complex, surpassing task-specific models (OptMATH) and optimization-aligned LLMs (ORLM). These results demonstrate our framework's ability to bridge general-purpose LLMs and domain-specific solvers through systematic reasoning decomposition.

**General Optimization Methods Comparison.** Table 2 shows our method excels in Execution Rate (ER), achieving near-perfect scores of 100% across benchmarks, including NL4Opt, Mamo Easy, Mamo Complex, NLP4LP, and ComplexOR. In Solution Accuracy (SA), it outperforms LL-MOPT on NLP4LP (87.6% vs. 83.8%) and OptiBench (69.3% vs. 66.4%), but slightly trails on ComplexOR (72.2% vs. 72.7%) and IndustryOR (38.0% vs. 44.0%). With a micro-average ER

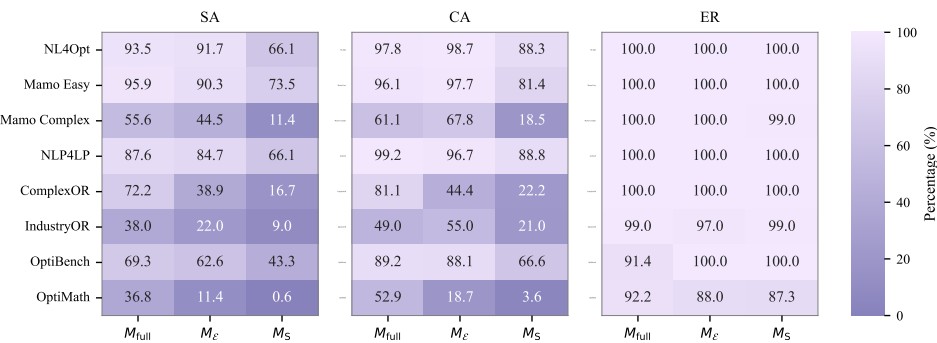

Figure 4: Heatmap of ablation study comparing $M_{\text{full}}$, $M_{\mathcal{E}}$, and $M_{\mathcal{S}}$ across benchmark datasets. Deeper purple indicates poorer performance.

of 97.0% and macro-average SA of 68.6%, our method demonstrates robust versatility, though improvements in specific SA metrics are needed.

Our proposed method showcases significant strengths in optimization problem modeling. It achieves state-of-the-art SA in multi-agent settings, with top scores on NL4Opt (93.5%), NLP4LP (87.6%), and ComplexOR (72.2%), alongside near-perfect ER scores of 100% across most benchmarks. In specific optimization tasks, it dominates on NL4Opt, Mamo Easy, and Mamo Complex, though it falls short on the Specific benchmark (37.4%) compared to DeepSeek-R1-671B (45.2%). For general optimization, its robust ER and competitive SA underscore its versatility. While it excels in many areas, enhancements are needed in specific metrics like IndustryOR and OptMATH, as well as broader SA performance, to further strengthen its capabilities.

### 3.3 Ablation Experiment

The primary ablation study validates the roles of the core architectural components: the extractor ($\mathcal{E}$) and the solver ($\mathcal{S}$). By comparing the full system $M_{\text{full}}$ (integrating $\mathcal{E}$ and $\mathcal{S}$), $M_{\mathcal{E}}$ (with $\mathcal{E}$ replaced by LLaMa3-70B), and $M_{\mathcal{S}}$ (with $\mathcal{S}$ replaced by LLaMa3-70B) across eight optimization benchmarks, we assess their contributions using SA, CA, and ER as metrics. As shown in Figure 4, $M_{\text{full}}$ significantly outperforms $M_{\mathcal{E}}$ and $M_{\mathcal{S}}$ across all benchmarks. For example, on the ComplexOR dataset, $M_{\text{full}}$ achieves a SA of 72.2%, whereas $M_{\mathcal{E}}$ and $M_{\mathcal{S}}$ only reach 45.6% and 51.1%, respectively. Similarly, the ER of $M_{\text{full}}$ remains near-perfect at 100.0%, compared to 68.9% for $M_{\mathcal{E}}$ and 77.8% for $M_{\mathcal{S}}$. This confirms the necessity of the synergistic integration of $\mathcal{E}$ and $\mathcal{S}$, with specialized components demonstrating significant advantages over general-purpose large models in optimization tasks.

To further dissect the sources of these performance gains, we conducted a detailed ablation study on the individual components of our composite reward function in Appendix D.1.3. This analysis reveals a clear hierarchy of their importance. The $R_{\text{format}}$ and $R_{\text{constr}}$ are foundational; their absence leads to structurally malformed text and logically flawed models, causing a catastrophic drop in performance across all benchmarks. The $R_{\text{sem}}$ is also critical, guiding the Extractor to identify the correct objective function. Without it, the model often solves a different problem entirely, rendering the final solution invalid. Finally, the $R_{\text{sim}}$ provides fine-grained semantic tuning by preventing subtle logical flaws, such as confusing variable or parameter identities. While its removal results in a less severe performance drop, it is crucial for achieving high accuracy. Collectively, these results confirm that each reward component addresses a unique and critical facet of the modeling challenge—from structural integrity to semantic accuracy—and that their synergy is fundamental to MURKA's high performance. Furthermore, we verified that our alignment process preserves the base model's general capabilities and tested the framework's scalability on smaller models, with detailed results presented in Appendix D.1.5.

To further enhance MURKA, we recommend improving $\mathcal{E}$'s information extraction by incorporating domain knowledge to boost CA in complex tasks and strengthening $\mathcal{S}$'s modeling capabilities by exploring advanced optimization algorithms to improve SA. Additionally, motivated by our reward component analysis which reveals a clear hierarchy of their importance, integrating dynamic reward weighting in the RL process could adaptively prioritize extraction accuracy for diverse problem

types. Future work could analyze the contributions of $\mathcal{E}$ and $\mathcal{S}$ across varying task complexities and optimize the framework for resource-constrained scenarios, potentially exploring lightweight model architectures for edge deployment.

## 4 Related Work

The research is positioned at the intersection of three active areas: synthetic data generation for optimization, multi-agent frameworks, and the application of reinforcement learning for LLM alignment.

**Data Synthesis for Optimization Problems.** High-quality data is crucial for training LLMs to handle OR tasks. To overcome the scarcity of real-world datasets, recent works have focused on synthetic data generation. Methods include semi-automated synthesis from industrial case studies [Huang et al., 2025], back-translation from mathematical expressions to natural language [Yang et al., 2024], and using rejection sampling to control problem complexity [Lu et al., 2025]. These approaches have been instrumental in creating large-scale datasets for supervised fine-tuning. Distinctly, MURKA leverages synthetic data not for direct supervision, but as an environment to train our Extractor agent via reinforcement learning, allowing it to learn robust policies from interactive feedback rather than static examples.

**Multi-Agent Frameworks.** Decomposing complex problem-solving into tasks for specialized agents has proven effective. In the OR domain, frameworks like Chain-of-Experts [Xiao et al., 2023] orchestrate agents with forward-solving and backward-error-correction mechanisms, while OptiMUS [AhmadiTeshnizi et al., 2024] utilizes modular collaboration to handle linear and mixed-integer programming. These frameworks typically coordinate agents through sophisticated prompt engineering and predefined workflows. MURKA builds on this collaborative paradigm but enhances agent expertise through direct training. Our agents are not just directed by prompts but are specialized via reinforcement learning and knowledge distillation, leading to more adaptive and capable collaboration.

**Reinforcement Learning for LLM Alignment.** Reinforcement learning is a powerful technique for aligning LLMs with complex objectives, moving beyond simple instruction following. While foundational methods like PPO [Schulman et al., 2017] and DPO [Rafailov et al., 2023] are widely used, recent advancements have focused on enhancing reasoning capabilities. Notably, group relative reward mechanisms, as employed in mathematical reasoning tasks [Shao et al., 2024, Guo et al., 2025], have shown success by optimizing policies based on the relative quality of multiple generated outputs. We adapt this paradigm to the structured domain of optimization. Our key contribution is the design of a composite reward function that provides granular feedback on multiple facets of the extraction taskincluding format, constraints, and semanticsthereby aligning the LLM specifically with the rigorous demands of mathematical optimization modeling.

## 5 Discussion

The MURKA framework presents significant practical implications for OR automation. By reducing expert dependency and computational costs, it enables rapid prototyping of optimization models across various industries. The released dataset and benchmark facilitate standardized evaluation of OR modeling capabilities in LLMs. Potential societal benefits include more accessible optimization tools for small businesses and accelerated decision-making in critical domains. However, several limitations remain: (1) MURKA assumes structured natural language problem descriptions, which may not fully capture real-world problem complexity. (2) While our combined reward mechanism improves extraction accuracy, it still relies on predefined templates that may require adaptation for novel problem types. (3) The knowledge distillation process depends on synthetic data quality, which could inherit biases from the teacher model.

## Acknowledgments

The research is partially supported by Innovation Program for Quantum Science and Technology 2021ZD0302900 and China National Natural Science Foundation with No.62132018, 62231015, "Pioneer" and "Leading Goose" R&D Program of Zhejiang, 2023C01029, and 2023C01143, Anhui Provincial Natural Science Foundation under Grant 2208085MF172 and the USTC Kunpeng-Ascend Scientific and Educational Innovation Excellence Center. We also thank Sijia Zhang, Shuli Zeng, Xiaotian Pan, and the anonymous reviewers for their comments and helpful feedback.

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

# A  Algorithm Symbol Definitions and Explanations

Table 3 lists all the key symbols used in the algorithm along with their explanations. These symbols cover various aspects of the algorithm, such as reward calculation, semantic analysis, and similarity checks. Each row in the table includes a symbol and its detailed explanation, ensuring that readers can accurately understand the meaning of each symbol. We recommend that readers refer to Table 3 as needed while reading the algorithm description to ensure their understanding of the algorithm is accurate and comprehensive.

Table 3: Algorithm Symbol Explanation

| Symbol | Explanation |
|---|---|
| $\omega_i$ | Weight coefficient for the $i$-th reward component |
| $r$ | Base reward for correct items in format and constraint checks |
| $r_1$ | Reward for numerical elements in semantic analysis |
| $r_2$ | Reward for defined variables/parameters in semantic analysis |
| $r_3$ | Penalty for undefined elements in semantic analysis |
| $\alpha$ | Penalty coefficient for redundant text length ($l_{\text{pre}}$, $l_{\text{post}}$) |
| $\beta$ | Penalty coefficient for set cardinality difference in similarity |
| $\gamma_1$ | Weight for variable similarity in reward calculation |
| $\gamma_2$ | Weight for parameter similarity in reward calculation |
| $\delta$ | Scaling factor for similarity reward |
| $R_{\text{min}}$ | Lower bound of reward value for boundary constraints |
| $R_{\text{max}}$ | Upper bound of reward value for boundary constraints |
| $p_{\text{strict}}$ | Strict matching mode for segmented structure validation |
| $p_{\text{soft}}$ | Lenient matching mode for label detection |
| $S$ | Set of key labels, e.g., $\{\text{Sets}, \text{Parameters}, \dots\}$ |
| $V$ | Set of variables from candidate responses or predefined |
| $P$ | Set of parameters from candidate responses or predefined |
| $E$ | Set of objective function elements for semantic analysis |
| $V_{\text{aim}}$ | Target variable set, predefined standard |
| $P_{\text{aim}}$ | Target parameter set, predefined standard |
| $V_{\text{act}}$ | Actual variable set from candidate responses |
| $P_{\text{act}}$ | Actual parameter set from candidate responses |
| $l_{\text{pre}}$ | Length of preceding redundant text before label |
| $l_{\text{post}}$ | Length of trailing redundant text after label |
| $\mathbb{I}(\cdot)$ | Indicator function: 1 if condition met, else 0 |
| $\text{clip}(x, a, b)$ | Clipping function, constrains $x$ to $[a, b]$ |
| $\mathcal{M}_{\text{emb}}$ | Embedding model, maps text to vector space |
| $\cos(\mathbf{u}, \mathbf{v})$ | Cosine similarity between vectors $\mathbf{u}$ and $\mathbf{v}$ |

# B  Algorithm Notations

## B.1  Structure Validation Reward Computation

Format validation stage (line 2–6): First, the integrity of the segmented structure is verified by strictly matching the pattern $p_{\text{strict}}$. Then, the existence of tags is detected using the loose pattern $p_{\text{soft}}$. Finally, a single-item reward $r$ is given for the accurate positioning of the key tag set $S = \{\text{Sets}, \text{Parameters}, \dots\}$. Meanwhile, a linear penalty $\alpha$ is imposed on the redundant text length ($l_{\text{pre}}$, $l_{\text{post}}$) in the tag context.

Constraint checking stage (line 7–14): As shown in line 7, this module first assigns a basic reward $r$ to the existing constraint segments. Then, it analyzes the constraint conditions one by one: when a legal operator is detected, $r$ is added (line 10); when a defined variable $v \in V \cup P$ is used, $r$ is added; when a non-negative constraint exists, an additional $r$ is added. Finally, the total value is constrained within a preset interval through the clip function, and stability control is completed as shown in line 14.

---

**Algorithm 2** Structure Validation Reward

---

**Require:** Candidate response $c_i$
**Ensure:** Format reward $R_{\text{format}}$, constraint reward $R_{\text{constr}}$
 1: Extract text content $t_i \leftarrow c_i.\text{content}$
 2: **Format Verification Phase**:
 3:    $R_{\text{strict}} \leftarrow r \cdot \mathbb{I}(\text{match}(p_{\text{strict}}, t_i))$
 4:    $R_{\text{soft}} \leftarrow r \cdot \mathbb{I}(\text{search}(p_{\text{soft}}, t_i))$
 5:    $R_{\text{xml}} \leftarrow \sum_{s \in S}[r \cdot \mathbb{I}(s \text{ exists with correct position})] - \alpha(l_{\text{pre}} + l_{\text{post}})$
 6:    $R_{\text{format}} \leftarrow R_{\text{strict}} + R_{\text{soft}} + R_{\text{xml}}$
 7: **Constraint Checking Phase**:
 8: Initialize $R_{\text{constr}} \leftarrow r \cdot \mathbb{I}(\text{Constraints section exists in } t_i)$
 9: **for** each constraint $k$ **do**
10:     $R_{\text{constr}} \leftarrow R_{\text{constr}} + r \cdot \mathbb{I}(\text{valid operator})$
11:     $R_{\text{constr}} \leftarrow R_{\text{constr}} + r \cdot \mathbb{I}(\text{uses defined variables})$
12:     $R_{\text{constr}} \leftarrow R_{\text{constr}} + r \cdot \mathbb{I}(\text{contains non-negative constraint})$
13: **end for**
14:     $R_{\text{constr}} \leftarrow \text{clip}(R_{\text{constr}}, R_{\text{min}}, R_{\text{max}})$
15: **return** $R_{\text{format}}, R_{\text{constr}}$

---

## B.2  Content Validation Reward Computation

Semantic analysis stage (line 2–5): After extracting the element set $E$ from the objective function, a hierarchical reward mechanism is implemented as shown in line 4: numerical elements receive a basic reward $r_1$, elements within the variable set $V$ or the parameter set $P$ receive a compliance reward $r_2$, and undefined elements trigger a penalty $r_3$. Finally, the reward value is ensured to be stable within the effective interval $[R_{\text{min}}, R_{\text{max}}]$ through the boundary constraint function.

Similarity checking stage (line 6–13): This module achieves semantic alignment of structured elements through embedding space measurement and imposes a linear penalty term on the cardinality difference between the variable set $V$ and the parameter set $P$. Then, the text is mapped to the $\mathbb{R}^d$ space through the embedding model $\mathcal{M}_{\text{emb}}$, and the bidirectional maximum cosine similarity between the target set and the candidate set is calculated. Finally, a weighted fusion strategy is adopted to convert the semantic similarity into a reward value, and the clip function is used to ensure numerical stability.

---

**Algorithm 3** Content Validation Reward

---

**Require:** Candidate response $c_i$
**Ensure:** Semantic reward $R_{\text{sem}}$, similarity reward $R_{\text{sim}}$
 1: Extract text content $t_i \leftarrow c_i.\text{content}$
 2: **Semantic Analysis Phase**:
 3: Extract variable set $V$, parameter set $P$, objective elements $E$
 4:    $R_{\text{sem}} \leftarrow \sum_{e \in E} \begin{cases} r_1 & \text{if } e \text{ is numerical} \\ r_2 & \text{if } e \in V \cup P \\ r_3 & \text{otherwise} \end{cases}$
 5:    $R_{\text{sem}} \leftarrow \text{clip}(R_{\text{sem}}, R_{\text{min}}, R_{\text{max}})$
 6: **Similarity Checking Phase**:
 7:    $V_{\text{act}} \leftarrow \text{ExtractXML}(t_i, \text{Variables})$
 8:    $P_{\text{act}} \leftarrow \text{ExtractXML}(t_i, \text{Parameters})$
 9:    $R_{\text{len}} \leftarrow -\beta(|V_{\text{aim}}| - |V_{\text{act}}| + |P_{\text{aim}}| - |P_{\text{act}}|)$
10:    $\text{sim}_V \leftarrow \frac{1}{|V_{\text{aim}}|} \sum_{v \in V_{\text{aim}}} \max_{u \in V_{\text{act}}} \cos(\mathcal{M}_{\text{emb}}(v), \mathcal{M}_{\text{emb}}(u))$
11:    $\text{sim}_P \leftarrow \frac{1}{|P_{\text{aim}}|} \sum_{p \in P_{\text{aim}}} \max_{q \in P_{\text{act}}} \cos(\mathcal{M}_{\text{emb}}(p), \mathcal{M}_{\text{emb}}(q))$
12:    $R_{\text{sim}} \leftarrow R_{\text{len}} + \delta(\gamma_1 \cdot \text{sim}_V + \gamma_2 \cdot \text{sim}_P)$
13:    $R_{\text{sim}} \leftarrow \text{clip}(R_{\text{sim}}, R_{\text{min}}, R_{\text{max}})$
14: **return** $R_{\text{sem}}, R_{\text{sim}}$

---

# C  Detailed Benchmark Descriptions

This appendix provides a detailed overview of the eight benchmarks used to evaluate our method: NL4Opt, Mamo Easy, Mamo Complex, NLP4LP, ComplexOR, IndustryOR, OptiBench, and Opt-MATH. These benchmarks collectively comprise 2,224 problem instances, covering over 20 real-world scenarios and eight optimization categories. The scenarios and optimization types for each benchmark are summarized in Table 10 and Table 11, respectively.

The optimization types are abbreviated as follows: LP denotes Linear Programming, IP denotes Integer Programming, MIP denotes Mixed Integer Programming, NP denotes Nonlinear Programming, CO denotes Combinatorial Optimization, MOP denotes Multi-objective Programming, and SOCP denotes Second-Order Cone Programming.

# D  Detailed Experimental Inventory

To comprehensively evaluate the MURKA framework, we elaborate in this section on the experimental design, adhering to the MIT License, Llama Community License, and Apache License 2.0 for all existing assets (code, models, datasets) used in this study to ensure full compliance with their terms of use.

## D.1  Experimental Design and Objectives Analysis

### D.1.1  Capabilities of Native Small-Scale LLMs in Optimization Tasks

Due to the high deployment costs of large-scale language models in practical scenarios, our preliminary work first investigates the capabilities of native small-scale LLMs in optimization tasks.

This experiment evaluates the performance of native small-scale LLMs on optimization tasks, comparing LLaMa3-8B and LLaMa3-70B on the NL4Opt and Mamo tasks. We employ Zero-shot, Few-shot, and CoT [Wei et al., 2022] prompting strategies, and compare these with a method that generates AMPL files followed by Gurobi solving. Results, shown in Table 4, indicate that both LLaMa3-8B and LLaMa3-70B perform poorly when directly handling high-precision optimization problems, highlighting the limitations of native small-scale LLMs in optimization tasks, particularly in scenarios requiring high precision. By generating AMPL files and solving with Gurobi, the optimization capabilities of LLaMa3-8B and LLaMa3-70B improve significantly, by 3.73Œ and 6.54Œ, respectively. The larger improvement in LLaMa3-70B suggests that larger-scale models have greater potential in optimization modeling and solving, likely due to their enhanced language understanding and generation abilities.

Inspired by Ramamonjison et al. [2022], we introduce an information extraction pipeline, using LLaMa3-8B for information extraction or expert-curated information, combined with LLaMa3-70B and Gurobi for modeling and solving. As shown in Figure 5, this approach improves CA by an average of 4.67% and SA by 7.37%, indicating that information extraction significantly enhances the model's understanding and modeling quality for optimization problems. Next work could explore optimizing automated information extraction to reduce reliance on expert-curated information.

Table 4: Comparison of the SA of the native small-scale LLMs in optimization tasks with and without Gurobi Solver.

| Model | Method | NL4Opt | Mamo Easy | Mamo Complex |
|---|---|---|---|---|
| LLaMa3-8B | Zero-shot | 0.0% | 0.3% | 0.0% |
| | Few-shot | 0.0% | 0.0% | 0.0% |
| | CoT | 0.8% | 0.9% | 0.0% |
| | Gurobi | 4.4% | 2.0% | 0.0% |
| LLaMa3-70B | Zero-shot | 1.6% | 8.3% | 3.8% |
| | Few-shot | 1.6% | 0.2% | 0.0% |
| | CoT | 1.6% | 9.8% | 1.9% |
| | Gurobi | 33.5% | 52.0% | 1.4% |

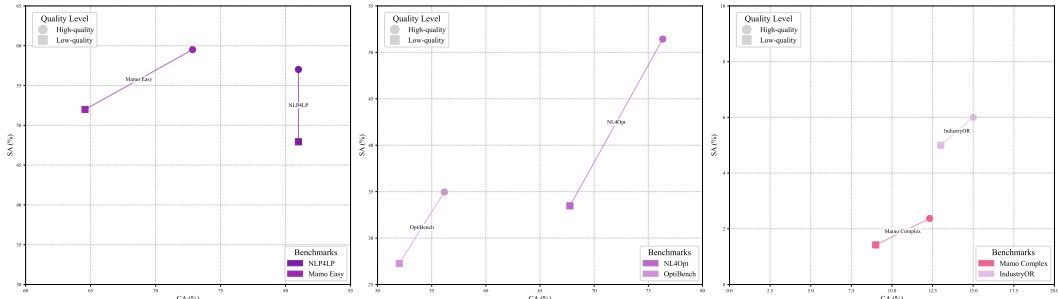

Figure 5: Effect of Extracted Information Quality on SA and CA.

### D.1.2 Performance Evaluation of MURKA Against Optimization Baselines

We propose the MURKA framework (see §2) to enhance the performance of large language models in optimization tasks. The Experimental Setup details the Benchmarks, Baselines, and Metrics used. Performance comparisons between MURKA and selected baselines are presented via radar charts in §3, with detailed data in Table 5 and Table 6. Results demonstrate that MURKA outperforms baselines across multiple optimization task benchmarks, exhibiting stronger robustness and generalization.

Table 5: Comparison of the SA metric against LLM-Assisted Methods.

| Method | NL4Opt | Mamo Easy | Mamo Complex | NLP4LP | Micro Avg |
|---|---|---|---|---|---|
| GPT-4 | 47.3% | 66.5% | 14.6% | 35.8% | 49.34% |
| Qwen3-32B | 83.5% | 70.7% | 27.0% | 76.0% | 57.68% |
| DeepSeek-R1-671B | 94.8% | 95.9% | 51.2% | 93.8% | 78.15% |
| Ours | 93.5% | 95.9% | **55.6%** | 87.6% | 76.48% |
| Method | ComplexOR | IndustryOR | OptiBench | OptMATH | Macro Avg |
| GPT-4 | 9.5% | 28.0% | 62.8% | 16.6% | 35.14% |
| Qwen3-32B | 33.3% | 21.0% | 56.0% | 13.86% | 47.67% |
| DeepSeek-R1-671B | 72.2% | 47.0% | 70.7% | 43.4% | 71.13% |
| Ours | **72.2%** | 38.0% | 69.3% | 36.8% | 68.61% |

Table 6: Comparison of the SA metric against Multi-Agent Methods.

| Method | NL4Opt | NLP4LP | ComplexOR | Micro Avg | Macro Avg |
|---|---|---|---|---|---|
| Reflexion | 53.0% | 46.3% | 19.1% | 48.45% | 39.47% |
| Chain-of-Experts | 64.2% | 53.1% | 38.1% | 57.76% | 51.80% |
| OptiMUS | 78.8% | 72.0% | 66.7% | 75.00% | 72.50% |
| Ours | **93.5%** | **87.6%** | **72.2%** | **89.80%** | **84.43%** |

### D.1.3 Detailed Ablation Studies on Key Components

As detailed in §3.3, our primary ablation study confirmed the critical, synergistic roles of the Extractor ($\mathcal{E}$) and Solver ($\mathcal{S}$). To further dissect the framework's performance gains, we conducted an additional ablation study on the individual components of our multi-part reward function.

Table 7: Ablation study on the impact of reward components on SA.

| Method | NL4Opt | NLP4LP | ComplexOR |
|--------|--------|--------|-----------|
| Ours (Full) | **93.5%** | **87.6%** | **72.2%** |
| w/o $R_{sim}$ | 78.6% | 73.4% | 58.5% |
| w/o $R_{sem}$ | 60.2% | 57.9% | 39.2% |
| w/o $R_{constr}$ | 55.8% | 52.1% | 34.6% |
| w/o $R_{format}$ | 51.3% | 48.5% | 30.1% |

The results, summarized in Table 7, reveal a clear hierarchy of importance among the reward components. The format reward is the most critical; its absence leads to malformed text and frequent compilation errors, severely undermining performance. The constraint reward and semantic reward are the next most critical, as their removal results in models that are either logically flawed due to incorrect constraints or solve an entirely different problem due to a wrong objective function. Finally, the similarity reward addresses more subtle logical errors, such as variable confusion; while its removal is less catastrophic, it remains essential for fine-tuning the model's accuracy.

### D.1.4    Generalization Assessment Across Diverse Tasks

This experiment evaluates the performance of MURKA-aligned LLaMa3-8B (Ours) against native LLaMa3-8B on seven tasks using the eval-harness [Gao et al., 2024], with accuracy as the metric (except for WMT14, which uses BLEU score). The MURKA-aligned model slightly outperforms native LLaMa3-8B on most tasks but shows minor declines on XNLI and WMT14. Overall, MURKA does not significantly degrade performance across a wide range of tasks.

Table 8: Performance comparison of MURKA and LLaMa3-8B across seven tasks, measuring accuracy (except WMT14, which uses BLEU score).

| Method | GSM8K | XNLI | TruthfulQA | ToxiGen | MMLU | QQP | WMT14 |
|--------|-------|------|-----------|---------|------|-----|-------|
| LLaMa3-8B | 76.48% | 44.76% | 54.09% | 43.19% | 68.00% | 42.80% | 36.81 |
| Ours | 78.86% | 43.95% | 54.56% | 43.40% | 68.01% | 48.76% | 36.52 |
| $\Delta$ | +2.38% | -0.81% | +0.47% | +0.21% | +0.01% | +5.96% | -0.29 |

### D.1.5    Evaluation of MURKA on Smaller-Scale Models

This experiment aligns LLaMa-8B and LLaMa-3B models using the MURKA framework and evaluates their performance on four benchmarks: NL4Opt, Mamo Easy, NLP4LP, and OptiBench, with ER and SA as metrics. Results, shown in Table 9, indicate that MURKA performs better on LLaMa-8B. Model scale has a greater impact on SA than ER, particularly in complex tasks. We recommend prioritizing larger models when resources permit to enhance performance.

Table 9: The SA and ER compared to the smaller-scale models.

| Metrics | ER | SA | ER | SA | ER | SA | ER | SA |
|---------|-----|-----|-----|-----|-----|-----|-----|-----|
| Method | NL4Opt | | Mamo Easy | | NLP4LP | | OptiBench | |
| LLaMa-8B | 100.0% | 93.5% | 100.0% | 95.9% | 100.0% | 87.6% | 91.4% | 69.3% |
| LLaMa-3B | 96.1% | 77.0% | 96.9% | 83.7% | 95.0% | 76.5% | 90.6% | 48.1% |
| $\Delta$ | -3.9% | -16.5% | -3.1% | -12.2% | -5.0% | -11.1% | -0.8% | -21.2% |

### D.2    Train Dataset Construction and Augmentation

To construct a high-quality dataset for optimization problems, we systematically curated data, ensuring a balanced representation of scenarios and problem types, as detailed in Table 10 and Table 11. From the test set, we randomly sampled 20% of the instances, stratified by scenarios and types, to serve as the foundation for knowledge distillation and data synthesis. Initially, we utilized

Table 10: The scenarios of the benchmarks.

| Scenarios | NL4Opt | Mamo Easy | Mamo Complex | IndustryOR | NLP4LP | ComplexOR | OptiBench | OptMATH |
|---|---|---|---|---|---|---|---|---|
| Agriculture | 17 | 30 | 5 | 6 | 14 | 0 | 56 | 0 |
| Energy | 5 | 33 | 7 | 1 | 5 | 0 | 22 | 6 |
| Health | 45 | 49 | 53 | 3 | 49 | 2 | 41 | 1 |
| Retail | 16 | 47 | 37 | 11 | 21 | 1 | 40 | 5 |
| Environment | 8 | 40 | 0 | 0 | 9 | 0 | 12 | 0 |
| Education | 3 | 32 | 0 | 3 | 3 | 0 | 9 | 0 |
| Financial Services | 8 | 46 | 2 | 6 | 6 | 0 | 21 | 6 |
| Transportation | 51 | 73 | 76 | 18 | 48 | 7 | 87 | 50 |
| Public Utilities | 4 | 29 | 11 | 0 | 4 | 1 | 18 | 12 |
| Manufacturing | 61 | 71 | 8 | 45 | 68 | 6 | 230 | 57 |
| Software | 1 | 0 | 10 | 1 | 1 | 1 | 5 | 2 |
| Construction | 3 | 56 | 1 | 1 | 3 | 0 | 26 | 0 |
| Legal | 0 | 0 | 0 | 0 | 0 | 0 | 0 | 0 |
| Customer Service | 0 | 2 | 0 | 0 | 1 | 0 | 3 | 0 |
| Entertainment | 4 | 44 | 0 | 0 | 6 | 0 | 6 | 0 |
| Others | 4 | 100 | 1 | 5 | 4 | 0 | 29 | 27 |
| Sum | 230 | 652 | 211 | 100 | 242 | 18 | 605 | 166 |

Table 11: The optimization types of the benchmarks.

| Types | NL4Opt | Mamo Easy | Mamo Complex | IndustryOR | NLP4LP | ComplexOR | OptiBench | OptMATH |
|---|---|---|---|---|---|---|---|---|
| LP | 98 | 2 | 59 | 20 | 56 | 6 | 422 | 14 |
| IP | 98 | 238 | 12 | 11 | 11 | 1 | 0 | 20 |
| MIP | 33 | 412 | 48 | 44 | 130 | 8 | 0 | 113 |
| NP | 0 | 0 | 2 | 0 | 26 | 0 | 183 | 9 |
| CO | 0 | 0 | 65 | 9 | 19 | 3 | 0 | 0 |
| MOP | 0 | 0 | 0 | 8 | 0 | 0 | 0 | 0 |
| SOCP | 0 | 0 | 0 | 0 | 0 | 0 | 0 | 10 |
| Others | 1 | 0 | 25 | 8 | 0 | 0 | 0 | 0 |
| Sum | 230 | 652 | 211 | 100 | 242 | 18 | 605 | 166 |

powerful inference models, such as DeepSeek-R1 and Qwen3-235B, as teacher models to generate seed data. These models performed reasoning-based analysis on the sampled problems, producing Problem-AMPL pairs, which map optimization problems to their AMPL formulations. To expand the diversity of scenarios, types, and data, we applied data augmentation techniques, leveraging prompt engineering and domain-expert input to refine and enrich the seed dataset while ensuring correctness of the AMPL formulations.

The augmented Problem-AMPL pairs were then solved using the Gurobi optimizer. We iteratively refined the AMPL models based on solver logs, adjusting formulations within a fixed number of iterations until successful convergence was achieved. Finally, information is extracted through prompt engineering and domain experts. Valid Problem-AMPL pairs were retained as the final output of this stage. Through this distillation and augmentation process, we generated a training dataset comprising 3,602 instances, ensuring complete separation between training and test sets to prevent data leakage. This dataset serves as a robust resource for training optimization models, with diverse scenarios and problem types reflective of real-world benchmarks.

### D.3 Computational Resources and Training Configuration

To solve the optimization model, we use Gurobi 12.0.1 [Gurobi Optimization, LLC, 2025]. The training of MURKA's extractor ($\mathcal{E}$) and solver ($\mathcal{S}$) components was conducted on a single GPU, leveraging its 24 GB of VRAM. The solver's longer training time and higher computational cost are attributed to its larger maximum sequence length and distinct hyperparameters, as detailed in Table 12. During inference, MURKA requires approximately 16 GB of VRAM (using FP16 precision), and the Gurobi solver runs on the CPU with a peak RAM usage of around 40 GB for the most complex cases.

When comparing computational costs, direct latency comparison with baselines like Chain-of-Experts is challenging, as they rely on API calls to closed-source models (e.g., GPT-4), making their performance subject to network variability and external load. Therefore, we advocate using token consumption as a more stable and equitable metric for cost, directly reflecting both API expenses and computational workload. Under this metric, MURKA is dramatically more efficient, reducing token usage by over 20× compared to Chain-of-Experts and also outperforming other learning-based methods. This provides strong, concrete evidence for the scalability and efficiency of our framework.

Table 12: Hyperparameter Configuration for Extractor and Solver.

| Model | Epoch | Batch | LearningRate | LoRA_R | Max_Length | WarmUp Ratio | Weight Decay | Adam Beta |
|---|---|---|---|---|---|---|---|---|
| $\mathcal{E}$ | 25 | 16 | 5e-6 | 16 | 1024 | 0.1 | 0.1 | 0.9 |
| $\mathcal{S}$ | 15 | 32 | 5e-5 | 8 | 3072 | 0 | 0 | 0.9 |

# E  Prompt template

## E.1  Information Extraction

**Prompt**

input_variables = ["question"],

template = """

You are a professional optimization problem analyst, proficient in extracting key elements from optimization problems described in natural language.
Your task is to accurately output the sets, parameters, variables, objective function, and constraints in a specific format.
Ensure that the output is concise, professional, and meets the requirements.

Here is the specific description of the optimization problem:
{question}

Please extract the required information from the following optimization problem according to the format below:

<Sets>
List the sets involved in the problem here. such as:
- set_name: description of the set
. . . . . .
</Sets>

<Parameters>
List the parameters involved in the problem here. such as:
- parameter_name: description of the parameter
. . . . . .
</Parameters>

<Variables>
List the variables involved in the problem here. such as:
- variable_name: description of the variable
. . . . . .
</Variables>

<Objective>
Specify the objective function of the problem here. such as:

```
  Maximize/Minimize: objective function
</Objective>

<Constraints>
List the constraints existing in the problem here. such as:
- constraint_1
......
</Constraints>

"""
```

## E.2 Code Generation

input_variables = ["question", "extracted_info"],

template = """

You are an optimization expert.

You should solve question step by step within specified label tags.

Solve the optimization problem and only output the complete AMPL model code within <AMPL></AMPL>tags.

Problem: {question}

Extract Information: {extracted_info}

Ensure that the output strictly adheres to the correct AMPL syntax and structure for model files.

"""

## E.3 Knowledge Distillation

input_variables = ["Problem"]

template = """

The following is an operations research problem.

Problem description: {Problem}

Let's solve it step by step:

step 1. Understand the problem

Please extract the key information for the optimization problem from the following natural language description:
- Problem description: Provide a detailed description of the task or problem, including the business background and the specific optimization goal.
- Decision variables: Identify the variables that need to be decided (e.g., quantity, time,

allocation, etc.).
- Objective function: Clearly define the optimization goal (e.g., maximizing profit, minimizing cost, minimizing time, etc.).
- Constraints: List all the limiting conditions that affect decision-making (e.g., resource constraints, time limitations, etc.).

step 2. Develop the mathematical model

Based on the analysis in step 1, construct the mathematical model:
- Decision variable symbols: Define each decision variable with a symbol and explain its meaning.
- Objective function: Express the optimization goal with a mathematical formula.
- Constraints: List all constraints and express them with mathematical formulas, including boundary conditions and other restrictions.

step 3. Implement the model in AMPL

Using the mathematical model developed in step 2, write the AMPL code:
- Declare variables: Define each decision variable and its value range.
- Objective function: Write the AMPL expression for the objective function.
- Constraints: Write the AMPL expressions for each constraint.

step 4. Output the AMPL code

Output the complete AMPL model code within <AMPL></AMPL>tags.

"""

