# OpenReview forum: "MURKA: Multi-Reward Reinforcement Learning with Knowledge Alignment for Optimization Tasks"
_NeurIPS.cc/2025/Conference — NeurIPS 2025 poster_

### Official Review · Reviewer_Pqa3 · 2025-07-01

**Clarity:** 2
**Significance:** 3
**Originality:** 3
**Rating:** 4
**Confidence:** 3

**Summary:**

The paper introduces MURKA, a reinforcement learning and knowledge distillation-based multi-agent framework for translating natural language descriptions of optimization problems into verifiable, executable solutions. MURKA leverages three specialized agents: Extractor, Solver and Checker. It is trained via a composite reward function and guided by distilled knowledge from a high-capacity teacher model. It achieves strong results on eight OR benchmarks, outperforming several state-of-the-art methods in both solution accuracy and execution robustness, despite using a relatively small 8B-parameter LLM backbone.

**Questions:**

* How does the composite reward function balance semantic accuracy vs syntactic format?
* What is the role of each reward component in the final performance gains?
* How sensitive is performance to the weights of the reward terms?

**Ethical Concerns:**

["NO or VERY MINOR ethics concerns only"]

**Final Justification:**

Thank you. I'll keep my score. During the rebuttal, my concern about the reliance on synthetic data has been resolved. However I still believe that the model underpeforms in certain domain-specific cases and the paper still remains difficult to read.

**Limitations:**

yes

**Quality:**

3

**Strengths And Weaknesses:**

Strengths:
* Novel architecture.
* Effective training.
* Competitive results.
* I like that it achieves high performance using a smaller LLaMa3-8B model, something that shows cost-efficiency.
* Thorough evaluation results and ablation study

Weaknesses:
* It relies on reliance on synthetic data, something that I don't like.
* Sometimes it slightly underperforms in certain domain-specific cases, for instance, in IndustryOR.
* Paper difficult to read, but the overall idea is good.

---

> ### Author Rebuttal · Authors · 2025-07-28
>
> We sincerely thank you for your detailed review and constructive feedback. We are encouraged that you recognized the novelty of our architecture, the effectiveness of our training strategy, the competitive results, and the cost-efficiency of achieving high performance with a smaller 8B model. We appreciate the opportunity to address your concerns regarding the reliance on synthetic data, performance on specific benchmarks, and the clarity of the paper. We hope the following clarifications will resolve your concerns.
>
> ### Point-by-Point Response
>
> #### **Question: How does the composite reward function balance semantic accuracy vs. syntactic format?**
>
> The balance is explicitly managed by the weighted combination of the four reward components, as defined in Equation 6. The weights ($ω_{format}$,$ω_{constr}$,$ω_{sem}$,$ω_{sim}$) were carefully tuned (as noted on line 163) to establish a hierarchy. **Syntactic format** ($R_{format}$) is foundational; an ill-formatted model is not executable. Therefore, its weight is prioritized to ensure the agent first learns to generate syntactically valid structures. **Semantic accuracy**, captured by the combination of $R_{sem}$, $R_{constr}$, and $R_{sim}$, is prioritized next. These rewards ensure the model is not only executable but also logically correct—representing the right objective function and constraints. This hierarchical weighting encourages the model to first become “fluent” in the AMPL language, and then “accurate” in its mathematical reasoning. Our ablation studies validate this design (see below).
>
> #### **Question: What is the role of each reward component in the final performance gains?**
>
> To directly answer this, we conducted an ablation study on the reward components. The results clearly demonstrate that each component plays a critical and distinct role in achieving the final performance.
>
> | **Method**      | **NL4Opt** | **NLP4LP** | **ComplexOR** |
> | --------------- | ---------- | ---------- | ------------- |
> | **Ours (Full)** | **93.5%**  | **87.6%**  | **72.2%**     |
> | $ω_{sim}=0$     | 78.6%      | 73.4%      | 58.5%         |
> | $ω_{sem}=0$     | 60.2%      | 57.9%      | 39.2%         |
> | $ω_{constr}=0$  | 55.8%      | 52.1%      | 34.6%         |
> | $ω_{format}=0$  | 51.3%      | 48.5%      | 30.1%         |
>
> *Table: Ablation study on reward components' impact on SA (%).*
>
>
> - **Removing Format Reward** ($R_{format}$): This is the most critical component. Without it, the Extractor produces malformed text, leading to frequent compilation errors in the downstream Solver. This severely impacts both execution and accuracy, especially on datasets with complex structure like ComplexOR.
> - **Removing Constraint Reward** ($R_{constr}$): This is the next most critical. Without it, the Extractor fails to correctly identify constraints, leading to models that are logically flawed or contain syntax errors. This results in a massive performance drop.
> - **Removing Semantic Reward** ($R_{sem}$): This is similarly vital. If the objective function is wrong, the model solves an entirely different problem. The impact is comparable to getting the constraints wrong, as it corrupts the core logic of the optimization model.
> - **Removing Similarity Reward** ($R_{sim}$): This reward addresses more subtle logical errors. Without it, the model might still produce a structurally sound model but could confuse variables (e.g., use the cost of Product A for Product B). This leads to significant, but less catastrophic, performance degradation compared to the other three components.
>
> #### **Question: How sensitive is performance to the weights of the reward terms?**
>
> Our hyperparameter tuning revealed a clear and logical hierarchy of sensitivity. The model's performance is most sensitive to the weights in the following order: $ω_{format} > ω_{constr} > ω_{sem} > ω_{sim}$. This ranking aligns perfectly with the ablation results above. A non-negotiable foundation of correct formatting is required before the logical components (constraints and objective) can have any meaning. The fine-grained similarity check provides a final polish. This robust hierarchy suggests that the reward design is principled. We will add a note on this sensitivity analysis to the appendix in the final version.
>
> #### **Concern: Reliance on synthetic data**
>
> We understand and share your concern about synthetic data.  Our primary motivation was to enable our reinforcement learning (RL) and knowledge distillation framework. **Large-scale, real-world datasets with structured annotations for this task are not readily available, particularly because optimization problems in industrial settings often involve sensitive business data, making them private and difficult to obtain for public research.** Therefore, synthetic data serves as a crucial and practical tool to distill reasoning patterns from a powerful teacher model and to provide the diverse feedback signals needed for our multi-reward RL process. Crucially, we demonstrate that this training strategy generalizes effectively to unseen, real-world problems. As shown in Tables 1 & 2 and Figure 3, MURKA achieves state-of-the-art or highly competitive performance across eight diverse, established benchmarks, indicating that the model learns robust, generalizable optimization principles rather than overfitting to synthetic artifacts. In the camera-ready version, we will add a more detailed discussion on this point in Section 5 (Discussion).
>
> #### **Concern: Slight underperformance in specific cases**
>
> We appreciate you pointing out this specific case. As shown in Table 2, our model (38.0% SA) slightly underperforms LLMOPT (44.0% SA) on the IndustryOR benchmark. We have analyzed our experiment logs and identified a key reason for this discrepancy. Due to hardware constraints during model alignment, we set a `Max_Length` of 3072 for the input sequence (as noted in Table 11). We found that many problems in the IndustryOR benchmark have context lengths that exceed this limit, leading to input truncation and a subsequent drop in performance. This is a technical limitation rather than a fundamental flaw in our method's reasoning capability. Despite this, we wish to emphasize that MURKA demonstrates superior overall performance and robustness. It outperforms all baselines on multiple other benchmarks (e.g., NL4Opt, NLP4LP) and achieves a significantly higher Execution Rate (ER) of 97.0% vs. 92.7% for LLMOPT, which is critical for real-world deployment. We will add a brief analysis of these specific failure cases to the appendix of the final paper.
>
> ### Concluding Remarks
>
> In summary, we have addressed the main concerns raised. We clarified that our use of synthetic data enables our framework and leads to strong generalization on real-world benchmarks, and we explained that the slight underperformance on IndustryOR stems from a technical sequence length limitation. Furthermore, our ablation study demonstrates the critical and distinct role of each component in our composite reward function. In response to your valuable feedback, we will revise the paper for clarity and add further analysis on data limitations, specific failure cases, and reward sensitivity to the appendix of the final version. We believe these clarifications and improvements will strengthen our paper and more clearly highlight our contribution.

---

> > ### Comment · Reviewer_Pqa3 · 2025-08-07
> >
> > Dear authors,
> >
> > Thank you for your thorough comment. I have no concern about the paper. I'll keep my positive score.
> >
> > Best.

---

### Official Review · Reviewer_jiyM · 2025-07-02

**Clarity:** 3
**Significance:** 3
**Originality:** 2
**Rating:** 4
**Confidence:** 4

**Summary:**

This paper introduces MURKA, a framework designed to automate the process of solving Operations Research problems from natural language descriptions. The core of MURKA is a multi-agent system composed of three specialized agents: an Extractor, a Solver, and a Checker. The Extractor, trained using reinforcement learning with a composite reward function, parses the natural language problem to identify key components. The Solver, enhanced through knowledge distillation from a powerful teacher model, then formulates this structured information into executable AMPL code. Finally, the Checker verifies the solution's correctness using feedback from an optimization engine like Gurobi. The authors demonstrate through extensive experiments on 2,224 OR problems that MURKA, using a LLaMa3-8B backbone, outperforms existing methods in solution accuracy and execution success rate.

**Questions:**

Questions
1. The composite reward function includes four weighted components.  How sensitive is the Extractor's performance to these weights, and how well would this reward function generalize to a new domain or problem type not represented in the training benchmarks (e.g., Second-Order Cone Programming)? Would introducing a new problem type require a full re-design and re-tuning of the reward function?
2. The shampoo example in Figure 1 illustrates the extraction of explicitly stated numerical constraints.  How does MURKA handle problems where constraints are implicit or require deeper semantic understanding and logical deduction to formulate correctly? For instance, a problem might describe relationships that need to be translated into a system of equations not explicitly provided in the text. How does the framework push the LLM to perform this kind of multi-step reasoning, rather than just information extraction?
3. What is AMPL? (A Mathematical Programming Language)?

**Ethical Concerns:**

["NO or VERY MINOR ethics concerns only"]

**Limitations:**

Yes, the authors have adequately addressed the limitations in Section 5.  They correctly identify the reliance on structured problem descriptions, the potential inflexibility of predefined reward templates, and the risk of inheriting biases from the teacher model during knowledge distillation.

**Paper Formatting Concerns:**

I have no concerns regarding the paper's formatting.

**Quality:**

3

**Strengths And Weaknesses:**

Strengths:
1. The paper tackles the challenging and highly relevant problem of automating OR modeling from natural language. The proposed MURKA framework is original in its combination of a multi-agent architecture, reinforcement learning with a multi-faceted reward function (GRPO), and knowledge distillation for this specific task. This approach presents a significant step forward from simpler prompt-based or supervised fine-tuning methods.
2. The design of the MURKA framework is well-structured. Decomposing the complex problem into specialized agent tasks (Extract, Solve, Check) is a logical approach that enhances modularity and interpretability.
3.  The paper is very well-written, clear, and easy to follow. The framework is explained logically, and Figure 2 provides an excellent visual overview of the different components (Extractor Alignment, Solver Alignment). The extensive appendices provide commendable detail on the experimental setup, hyperparameters, and prompts, ensuring a high degree of reproducibility.

Weaknesses:
1. The framework's effectiveness, particularly the Solver agent, is heavily dependent on knowledge distillation from a high-capacity teacher model like DeepSeek-R1. This creates a potential bottleneck, as the student model's quality is capped by the teacher's performance, and it requires access to powerful, often proprietary models for training. This reliance is particularly critical because, as the paper's own results show, even the best-performing LLMs struggle with the most complex OR problems; solution accuracy on the Mamo Complex and Specific benchmarks remains low, around 51-55% and 37-45% respectively. This highlights a more fundamental issue: while the framework excels at teaching an LLM to structure explicitly stated information into a correct format (like AMPL code), its ability to infer complex, unstated constraints or formulate entirely new mathematical relationships seems limited. The framework serves more as a powerful method for optimizing an LLM's performance on known problem patterns rather than a system that fundamentally expands its core logical and mathematical reasoning abilities for the OR domain.

---

> ### Author Rebuttal · Authors · 2025-07-28
>
> We sincerely thank you for your thorough evaluation, constructive feedback, and insightful questions. We are encouraged that you found our MURKA framework to be original, well-structured, and the paper clear and reproducible. We will first address your technical questions before turning to the concerns about model performance and reasoning depth, aiming to comprehensively clarify all points raised.
>
> ### Point-by-Point Response
>
> #### **Question: Sensitivity and generalization of the composite reward function.**
>
> 1. **Sensitivity to Weights**: To analyze sensitivity, we conducted an ablation study in which each reward component was removed in turn to assess its individual contribution. Results are summarized below:
>
> | **Method**      | **NL4Opt** | **NLP4LP** | **ComplexOR** |
> | --------------- | ---------- | ---------- | ------------- |
> | **Ours (Full)** | **93.5%**  | **87.6%**  | **72.2%**     |
> | $ω_{sim}=0$     | 78.6%      | 73.4%      | 58.5%         |
> | $ω_{sem}=0$     | 60.2%      | 57.9%      | 39.2%         |
> | $ω_{constr}=0$  | 55.8%      | 52.1%      | 34.6%         |
> | $ω_{format}=0$  | 51.3%      | 48.5%      | 30.1%         |
>
> *Table: Ablation study on reward components' impact on SA (%).*
>
> These results demonstrate that each component is essential. The dramatic drop in performance upon the removal of any single component robustly validates the synergistic effect and soundness of our multi-faceted reward mechanism. In particular, the catastrophic failure when the format ($ω_{format}$) and core logic ($ω_{constr}$, $ω_{sem}$) rewards are removed underscores that the model must first grasp the fundamental structure of a problem to have any chance of solving it effectively.
>
>
> 2. **Generalization**: Our reward design is based on general principles of mathematical optimization—extracting variables, parameters, objectives, and constraints—rather than task-specific heuristics. This abstraction allows MURKA to generalize across diverse optimization types, including MIP, LP, and NLP. As shown in Table 5, MURKA achieves consistent results across seven different optimization categories.  For a new optimization type like Second-Order Cone Programming (SOCP), which can be expressed with these same fundamental components, the reward function would generalize well without requiring a full redesign. For tasks outside optimization, we show in Table 9 of our paper that our alignment does not significantly degrade the model's general capabilities.
>
> We will incorporate this ablation study and discussion into the final version of the paper.
>
> #### **Question: What is AMPL?**
>
> Thank you for catching this omission. We will revise the paper to define AMPL (A Mathematical Programming Language) upon first mention, as referenced in line 79 ([Fourer et al., 2003]).
>
> To elaborate, AMPL is a high-level, algebraic modeling language designed specifically for describing and solving large-scale optimization problems. We chose AMPL for several key reasons that align with our framework's design:
>
> - **Separation of Model and Data**: AMPL allows the logical structure of a problem (the model) to be defined separately from the specific data used. This modularity is a perfect fit for our Extractor-Solver architecture. The Solver agent can focus entirely on generating a syntactically and logically correct model file, while the specific numerical values (parameters) can be handled independently.
> - **Solver Independence**: An AMPL model can be solved by a wide variety of optimization engines (e.g., Gurobi) without changing the model itself. This provides flexibility and allows us to leverage the power of state-of-the-art commercial solvers like Gurobi for the final numerical optimization step.
> - **Readability**: AMPL's syntax closely resembles standard mathematical notation, making the generated models relatively easy for human experts to read, verify, and debug, which is crucial for building trust in automated systems.
>
> We will clarify this in the final version of the manuscript to better justify our choice of AMPL as the target language.
>
> #### **Concern: Dependency on a powerful teacher model and the performance cap on complex problems.**
>
> Thank you for raising this critical point, which touches upon both the practical value of our work and a fundamental challenge for the field. Our response is twofold:
>
> 1. **Value Proposition: Enabling Practical, Low-Latency Deployment**: We agree that the student model's performance is influenced by the teacher. However, the primary goal of our knowledge distillation strategy (Sec 2.3) is not simply to mimic the teacher, but to **democratize and operationalize** its capabilities. We transfer the specialized reasoning patterns of a massive, proprietary model like DeepSeek-R1 (a 671B parameter model) into a much smaller, more efficient, and openly accessible student model (LLaMa3-8B).
>
>    This addresses a critical barrier to real-world adoption. Many industrial applications, such as dynamic production scheduling on a factory floor or real-time logistics routing, are highly sensitive to network latency and require local, on-premise deployment. Relying on constant API calls to a massive, cloud-hosted model is often infeasible due to network delays, data privacy concerns, and prohibitive operational costs. By successfully distilling expertise into a compact 8B model, MURKA enables the development of lightweight, locally deployable OR tools, which we see as its primary application value. The fact that our 8B model approaches the performance of a model over 80x larger (Fig 3a) highlights the success of this practical knowledge transfer.
>
> 2. **Performance on Complex Problems as a Field-Wide Challenge**: We fully agree with your insightful observation that even powerful models like DeepSeek-R1 struggle with the most complex OR problems. This indeed suggests, as you noted, that current LLMs may be adept at generating syntactically correct formats without achieving full, deep logical comprehension. This is an important, acknowledged limitation of the *current generation of all LLMs*, not a flaw unique to our framework.
>
>    Our results should be viewed in this context. As shown in Table 1 of our paper, DeepSeek-R1 achieves only 51.2% on Mamo Complex. Our 8B model's performance (55.6%) is not only highly competitive but even surpasses the teacher in this instance. This demonstrates that while the absolute performance ceiling is tied to the progress of foundation models, our framework provides a state-of-the-art method for aligning models to this complex domain, making the most of the reasoning capabilities that are currently available.
>
> #### **Concern: The framework is more for formatting than deep reasoning and struggles with implicit constraints.**
>
> We apologize if the shampoo example in Figure 1 oversimplified our framework's capabilities. We agree that natural language descriptions are often ambiguous. A core, implicit benefit of our framework is that it forces a translation from this ambiguous prose into a structured, formal representation. This process of extraction and structured code generation inherently acts as a disambiguation step, clarifying the problem's logical relationships before solving.
>
> We will add a more complex example to the appendix in the final version to better illustrate this process.
>
> ### Concluding Remarks
>
> In summary, we have addressed the key points raised in your review. We demonstrated the necessity of each component in our composite reward function with a new ablation study and clarified that its design is generalizable. We also elaborated on our choice of AMPL, highlighting its suitability for our framework. Furthermore, we contextualized our work's primary contribution as enabling practical, low-latency OR modeling. We believe these clarifications and additions will significantly strengthen the paper and are grateful for the opportunity to improve our work.

---

> > ### Author Response · Authors · 2025-08-08
> >
> > Dear Reviewer,
> >
> > I hope this message finds you well. As the discussion period is nearing its end with less than two days remaining, I wanted to ensure we have addressed all your concerns satisfactorily. If there are any additional points or feedback you'd like us to consider, please let us know. Your insights are invaluable to us, and we're eager to address any remaining issues to improve our work.
> >
> > Thank you for your time and effort in reviewing our paper.

---

> > ### Comment · Reviewer_jiyM · 2025-08-08
> > **Make the rebuttal short and precise**
> >
> > The authors address the questions proposed in the original review. But a short and precise version is better to grasp the key points. I will keep the score.

---

### Official Review · Reviewer_DipB · 2025-07-03

**Clarity:** 3
**Significance:** 3
**Originality:** 3
**Rating:** 5
**Confidence:** 2

**Summary:**

The paper presents MURKA, an end-to-end framework that automates converting natural-language optimization problems into executable models and verified solutions via a three-agent pipeline including Extractor, Solver and Checker.​

Key Contributions​
* Modular design separates extraction, generation, and verification, boosting transparency and reducing latency.​
* Multi-dimensional RL reward offers fine-grained feedback on extracted specs' syntax and semantics.​
* Cross-domain knowledge distillation lets the 8B-parameter model match or outperform larger LLMs in optimization tasks at lower compute costs.​
* Validated on 8 benchmarks (e.g., NL4Opt, Mamo), with 5.9% higher solution accuracy and 5.1% higher execution success rate than prior methods. Ablation studies confirm Extractor and Solver value.​

MURKA is an effective, scalable alternative to prompt engineering or monolithic fine-tuning for LLM-driven optimization.

**Questions:**

1. Could you please provide more analysis on the reason why the performance of the proposed method is worse than DeepSeek R1? Is it because long-CoT models with enhanced reasoning capabilities are especially useful in this task? Is it indicate that R1-distilled qwen 7B should be used in this task rather than llama3 8B?

**Ethical Concerns:**

["NO or VERY MINOR ethics concerns only"]

**Quality:**

3

**Strengths And Weaknesses:**

​One of the key aspects of MURKA is its modular design. By clearly separating the tasks of information extraction, model generation, and verification, the framework enhances transparency. This modularity also has the added benefit of reducing latency, as each component can be optimized independently. The ability to understand and modify individual parts of the system without affecting the others makes MURKA more accessible and easier to maintain.​

The multi-dimensional RL reward system employed in MURKA provides fine-grained feedback on both the syntactic and semantic aspects of the extracted specifications. This detailed feedback mechanism is essential for improving the quality of the information processed by the framework. It allows for more precise adjustments during the learning process, ensuring that the output models are both accurate and meaningful.​

Another significant feature of MURKA is its use of cross-domain knowledge distillation. This technique enables an 8B model to match or even outperform larger models in optimization tasks while keeping compute costs low. By leveraging knowledge distillation, MURKA can achieve high-performance results with a relatively smaller model, making it more resource-efficient and scalable.​

However, one notable concern regarding MURKA is its performance in comparison to DeepSeek R1. Although MURKA shows strong results on some tasks, there are instances where its performance falls short of that of DeepSeek R1. It is possible that long-CoT models with enhanced reasoning capabilities, such as DeepSeek R1, have an inherent advantage in NL-to-optimization tasks. The performance gap might indicate that for achieving optimal results, alternative models like R1-distilled Qwen 7B, which could potentially combine the benefits of knowledge distillation with more advanced reasoning mechanisms, might be more appropriate than the LLaMA3 8B model used in MURKA.

---

> ### Author Rebuttal · Authors · 2025-07-28
>
> We sincerely thank you for your thoughtful and encouraging review. We especially appreciate your recognition of our key contributions, including the transparency of the modular design, the fine-grained feedback from the multi-dimensional RL reward, and the efficiency of our cross-domain knowledge distillation. We appreciate your insightful question regarding the performance comparison with DeepSeek-R1, and we hope the following clarifications will address your concern and reinforce the value of our work.
>
> ### Point-by-Point Response
>
> #### **Concern about performance compared to DeepSeek-R1.**
>
> The comparison to DeepSeek-R1 is indeed a crucial point. As noted in Table 7, the baseline DeepSeek-R1 used in our comparison is a **671B-parameter model**, which is over **80× larger** than the LLaMA3-8B model used in MURKA. Despite this disparity, MURKA achieves comparable or even superior performance on several benchmarks (e.g., Mamo Complex: 55.6% vs. 51.2%), underscoring our framework’s effectiveness and computational efficiency. This is a primary contribution of our work: bridging the performance gap with significantly smaller, more practical, and deployable models.
>
> We agree with your hypothesis that models with enhanced reasoning capabilities are particularly effective for NL-to-optimization tasks. Your insight that performance is tied to the problem's characteristics is excellent and gets to the heart of the matter. We hypothesize the performance differences stem from two distinct types of challenges in NL-to-Optimization:
>
> - **Complex Reasoning Chains:** For problems where the solution requires intricate, multi-step logical deductions and dependencies (e.g., complex conditional logic), a model with strong Long-CoT capabilities like DeepSeek-R1 naturally has an advantage. Its general reasoning prowess allows it to connect disparate pieces of information across a long context. This is likely why it excels on certain benchmarks.
> - **Robust Structural Extraction:** However, many optimization problems hinge on the precise and robust extraction of numerical values, relationships, and constraints from dense or ambiguously worded text. This is where MURKA excels. A generic Long-CoT approach can sometimes be brittle, misinterpreting a single critical value or relationship, leading to an entirely incorrect formulation. Our **Extractor agent**, in contrast, is specifically optimized via a multi-dimensional RL reward to systematically and accurately parse these structural components. It is trained to be robust against the very ambiguities that can derail a general reasoning process.
>
> In essence, while DeepSeek-R1's strength lies in its general problem-solving, MURKA's advantage is in its specialized, high-fidelity conversion of natural language into a structured, verifiable model.
>
> #### **Concern about the choice of the LLaMA3-8B backbone.**
>
> You raise an excellent point about our choice of LLaMA3-8B versus other models like Qwen-7B. Our selection of LLaMA3-8B was based on its strong, publicly verifiable performance in English comprehension—as evidenced by its leading position on the Hugging Face Open LLM Leaderboard for models of its class, making it a solid foundation for our experiments. However, the core technical contribution of our work is the **model-agnostic nature of the MURKA framework itself**. Our framework is designed as a plug-and-play enhancement for any capable base model. It can be seamlessly applied to Qwen-7B, LLaMA3-8B, or any future, more advanced open-source models. This adaptability is key because, as you rightly suggest, the ideal solution involves a synergy between different capabilities.
>
> ### Concluding Remarks
>
> In summary, we have addressed your valuable questions by clarifying our design choices. We explained that MURKA's competitive performance against much larger models stems from its specialized strength in **robust structural extraction**, which complements the general reasoning abilities of models like DeepSeek-R1 by targeting a different, yet critical, set of problem characteristics. Furthermore, we highlighted that the MURKA framework is intentionally **model-agnostic**, designed to synergize with and enhance any underlying LLM. Based on this valuable discussion, we will expand the future work section in the camera-ready version to explicitly discuss these complementary strengths and the framework's adaptability. We believe these clarifications strengthen the paper and are grateful for the opportunity to discuss these points.

---

### Official Review · Reviewer_LeaD · 2025-07-03

**Clarity:** 2
**Significance:** 2
**Originality:** 2
**Rating:** 4
**Confidence:** 2

**Summary:**

This paper presents MURKA, a framework that uses reinforcement learning and knowledge distillation to convert natural language descriptions into optimization programs. The system has three agents: an Extractor that parses problem descriptions, a Solver that generates AMPL code, and a Checker that verifies solutions. The Extractor is trained with a modified GRPO algorithm using multi-dimensional rewards. The Solver learns from a teacher model through knowledge distillation. Tests on eight benchmarks show MURKA achieves 5.9% better solution accuracy and 5.1% better execution success compared to existing methods.

**Questions:**

Can you provide concrete numbers on training time, inference latency, and memory usage? How does MURKA compare to baselines in terms of computational cost?

**Ethical Concerns:**

["NO or VERY MINOR ethics concerns only"]

**Limitations:**

The authors partially address limitations in Section 5. They mention assumptions about structured input, template dependency, and potential bias from teacher models. However, they don't discuss computational costs, and all that ., so I would recommend the authors to improve this section.

**Quality:**

2

**Strengths And Weaknesses:**

1.nThe paper presents a well-structured pipeline with three specialized agents. The division of labor between extraction, solving, and checking makes sense.
2. The authors test on eight different benchmarks covering various optimization problems. Results show consistent improvements across most metrics.
3.The authors show that both the Extractor and Solver components contribute to performance, with clear evidence in Figure 4.

4.The core ideas (multi-agent systems, RL for LLMs, knowledge distillation) are not new. The contribution is mainly in combining these existing techniques.
5. While the paper claims to address scalability (C1), there's no analysis of training time, inference latency, or computational requirements compared to baselines.

---

> ### Author Rebuttal · Authors · 2025-07-27
>
> We sincerely thank you for your time and for providing thoughtful and constructive feedback on our work. We are encouraged by the recognition of our well-structured pipeline, extensive experiments evaluation, and clear ablation studies clear.
> We appreciate the critical points raised, especially regarding computational cost analysis and novelty. Below, we address each concern in detail and outline the improvements we will incorporate in the final version.
>
> ### Point-by-Point Response
>
> #### **Concern: Lack of computational cost analysis**
>
> We appreciate the reviewer’s request for a more detailed computational cost analysis. While we initially provided this in Appendix D.2 and D.3, we agree that these insights should be brought forward and clarified. Below, we offer a comprehensive summary:
>
> **Training Cost & Data Efficiency:** Our framework is highly efficient in terms of both training resources and data requirements.
>
> - **Hardware and Time:** The Extractor and Solver were trained on a single NVIDIA RTX 4090 GPU (24GB). Training times were approximately 46 hours for the Extractor and 110 hours for the Solver (see Appendix D.3, Table 11).
> - **Data Efficiency:** Crucially, MURKA achieves superior performance while being remarkably data-efficient. Our training dataset contains only 3,602 instances, which is merely **36.7%** of the data used by a strong baseline like LLMOPT.
>
> Despite using significantly less training data, MURKA surpasses LLMOPT's performance, highlighting the effectiveness and efficiency of our alignment strategy. This proves our approach not only achieves state-of-the-art results but does so with substantially lower data and computational overhead for training.
>
> **Inference Cost:**
>
> - **Latency:** In terms of computational efficiency, our method exhibits superior performance. On the NL4Opt, NLP4LP, and ComplexOR benchmarks, MURKA’s average total inference latencies (the sum of LLM generation time and Gurobi solver time) are 37.0s, 36.1s, and 54.5s respectively. This represents a speedup of over 2x compared to OptiMUS, confirming a substantial advantage in efficiency.
> - **Memory:** During inference, MURKA requires approximately 16 GB of VRAM (using FP16 precision). The Gurobi solver runs on CPU with a peak RAM usage of around 40 GB for the most complex cases.
>
> **Cost Comparison with Baselines:** Direct latency comparison with baselines like Chain-of-Experts is challenging, as they rely on API calls to closed-source models (e.g., GPT-4), making their performance subject to network variability and external load. Therefore, we advocate using token consumption as a more stable and equitable metric for computational cost, as it directly reflects both API expenses and computational workload.
>
> | **Method**       | **NL4Opt** | **NLP4LP** | **ComplexOR** |
> | ---------------- | ---------- | ---------- | ------------- |
> | Chain-of-Experts | 38230.2    | 42985.2    | 47740.1       |
> | OptiMUS          | 15937.4    | 12972.0    | 13522.2       |
> | LLMOPT           | 1912.0     | 2078.3     | 4530.7        |
> | **Ours**         | **1602.2** | **1621.0** | **2067.8**    |
>
> *Table: Average Token Count (Prompt + Completion) per Task*
>
> As the table above clearly shows, MURKA is **dramatically more efficient**, reducing token usage by over 20x compared to Chain-of-Experts and also outperforming other learning-based methods. This provides strong, concrete evidence for our claim of creating a scalable framework (C1).
>
> #### **Concern: Novelty of the work**
>
> We greatly appreciate you raising the concern about novelty, as it gives us an opportunity to more clearly articulate the core contribution of our work. As you pointed out in the Strengths section, the true novelty of our work lies in the systematic and organic integration of these existing techniques, along with their targeted adaptation, to create a "well-structured" and collaborative framework where the "division of labor makes sense." This framework is designed to efficiently solve the complex challenge of converting natural language into verifiable optimization programs.
>
> ### Concluding Remarks
>
> In summary, we hope the responses above have clarified the two main issues you raised. First, we have provided a detailed computational cost analysis, demonstrating that the MURKA framework holds significant advantages in training data efficiency, inference latency, and token consumption. Second, we have reiterated that the core novelty of our work lies in the systematic integration and targeted adaptation of existing techniques to create an efficient, collaborative framework. Based on your valuable feedback, we will incorporate the full cost analysis into the final version of our paper and restructure the introduction to better highlight the systemic innovation of our work.

---

### Note · Authors · 2025-08-12

Dear Area Chair and Reviewers,

We sincerely thank all reviewers for their constructive feedback and for their uniformly positive assessment of our work. We are grateful for the productive discussion period and pleased that our rebuttal successfully addressed all initial concerns, with no new issues raised.

We are encouraged that reviewers found our framework "well-structured," which reflects our core contribution: the systematic integration of multi-agent RL and knowledge distillation. This approach advances the automation of optimization modeling by achieving state-of-the-art results with practical, low-cost deployability.

We believe these contributions advance the field of LLM-driven optimization and appreciate the opportunity to share our work with the NeurIPS community.

---

### Decision · Program_Chairs · 2025-09-17

**Decision:**

Accept (poster)

**Comment:**

This paper presents MURKA, a multi-agent framework combining reinforcement learning with a composite multi-dimensional reward and knowledge distillation to translate natural language descriptions of optimization problems into executable, verifiable programs. The modular Extractor–Solver–Checker design improves interpretability and efficiency, with ablation studies showing each component’s necessity. Experiments on eight OR benchmarks demonstrate consistent gains in solution accuracy and execution success over baselines, while achieving low inference cost. Reviewers praised the technical soundness, thorough evaluation, and practical deployability, though they noted moderate conceptual novelty, reliance on synthetic data, and limited performance on the most complex reasoning tasks. The authors’ rebuttal addressed these points with additional cost analysis and clarifications. However, the writing requires significant improvement: the paper is at times difficult to follow and not fully self-contained, which reduces accessibility for the broader community. Taking all factors into account, I find the contributions meaningful and empirically validated, but with moderate novelty and clarity issues. I recommend acceptance as a Poster, but strongly urge the authors to substantially improve clarity and self-containedness in the camera-ready version.